# Oncolytic Adenovirus Armed with a Novel Agonist of the CD137 Immune Checkpoint Stimulator Suppresses Tumor Growth

**DOI:** 10.3390/vaccines12030340

**Published:** 2024-03-21

**Authors:** Martin R. Ramos-Gonzalez, Mohammad Tarique, Lalit Batra, Feyza Arguc, Rodolfo Garza-Morales, Haval Shirwan, Esma S. Yolcu, Jorge G. Gomez-Gutierrez

**Affiliations:** 1Roy Blunt NextGen Precision Health Building, University of Missouri, Columbia, MO 65211, USA; mr.ramos@health.missouri.edu (M.R.R.-G.); mohammad.tarique@health.missouri.edu (M.T.); f.arguc@health.missouri.edu (F.A.); haval.shirwan@health.missouri.edu (H.S.); esma.yolcu@health.missouri.edu (E.S.Y.); 2Ellis Fischel Cancer Center, University of Missouri, Columbia, MO 65212, USA; 3Regional Biocontainment Laboratory, Center for Predictive Medicine, University of Louisville, Louisville, KY 40222, USA; lalit.batra@louisville.edu; 4Division of Hematology and Medical Oncology, Mayo Clinic, Phoenix, AZ 85054, USA; garzamorales.rodolfo@mayo.edu; 5Department of Pediatrics, University of Missouri, Columbia, MO 65212, USA

**Keywords:** oncolytic, adenovirus, SA-4-1BBL, lung, cancer, immunotherapy

## Abstract

Natural 4-1BBL (CD137L) is a cell membrane-bound protein critical to the expansion, effector function, and survival of CD8^+^ T cells. We reported the generation of an active soluble oligomeric construct, SA-4-1BBL, with demonstrated immunoprevention and immunotherapeutic efficacy in various mouse tumor models. Herein, we developed an oncolytic adenovirus (OAd) for the delivery and expression of SA-4-1BBL (OAdSA-4-1BBL) into solid tumors for immunotherapy. SA-4-1BBL protein expressed by this construct produced T-cell proliferation in vitro. OAdSA-4-1BBL decreased cell viability in two mouse lung cancer cell lines, TC-1 and CMT64, but not in the non-cancerous lung MM14.Lu cell line. OAdSA-4-1BBL induced programmed cell death types I and II (apoptosis and autophagy, respectively), and autophagy-mediated adenosine triphosphate (ATP) release was also detected. Intratumoral injection of OAdSA-4-1BBL efficiently expressed the SA-4-1BBL protein in the tumors, resulting in significant tumor suppression in a syngeneic subcutaneous TC-1 mouse lung cancer model. Tumor suppression was associated with a higher frequency of dendritic cells and an increased infiltration of cytotoxic CD8^+^ T and NK cells into the tumors. Our data suggest that OAdSA-4-1BBL may present an efficacious alternative therapeutic strategy against lung cancer as a standalone construct or in combination with other immunotherapeutic modalities, such as immune checkpoint inhibitors.

## 1. Introduction

4-1BBL (CD137L) is as a costimulatory molecule required for appropriate T-cell expansion, inducing the effector and memory responses in CD8^+^ T cells, a mechanism required for the immune system to fight tumoral cells [1]. Endogenous 4-1BBL is functional only a transmembrane protein, but not in soluble form [2]. To overcome this limitation, our group previously reported the generation of a novel active form by linking the extracellular domain of murine 4-1BBL with the streptavidin (SA) core to allow formation of soluble active oligomers. The resulting protein SA-4-1BBL has pleiotropic effects on cells of innate, adaptative and regulatory immunity with therapeutic efficacy in various preclinical model [3,4]. In a recent study, we demonstrated tumor immunoprevention efficacy of SA-4-1BBL as a single agent mediated by CD4^+^ T and NK cells [5]. These findings are promising for the study of the antitumoral capacities of SA-4-1BBL as a coadjuvant in clinical settings. However, the production of this protein can be time- and cost-intensive, limiting the access to this new therapy. For this reason, we developed a new platform that can carry the genetic sequence and induce the production and release of the SA-4-1BBL protein by incorporating it into the genome of a novel oncolytic viral construct that can function as a second therapeutic agent by directly injecting it into the tumor or by respiratory delivery to target lung cancer.

Oncolytic virotherapy is an emerging anti-tumor therapeutic approach that uses cancer selective tumoricidal agents, which replicate in and destroy tumor cells while leaving normal cells undamaged [6]. Oncolytic adenoviruses (OAds) are an optimal choice to express and deliver SA-4-1BBL, as they not only kill cancer cells directly, but also prime the host’s antitumor immunity and effectively deliver immunostimulatory signals preferentially within the tumor milieu (Figure 1).

Oncolytic viruses are also immunogenic as they can stimulate interferon (IFN)-type I responses in antigen-presenting cells (APCs) by binding to Toll-like receptors (TLRs) [7]. More importantly, OAd infection can activate immunogenic cell death (ICD) in cancer cells by modifying the tumor microenvironment (TME), inducing dying cells to release proinflammatory cytokines, tumor-associated antigens (TAAs), damage-associated molecular pattern molecules (DAMPs) and pathogen-associated molecular pattern molecules (PAMPs), enhancing the recruitment of activated immune cells [8,9]. However, the use of oncolytic virus in clinical trials has shown limited efficacy when used as a monotherapy [10,11]. To increase the therapeutic potential beyond oncolysis induction, several modifications have been made to the fiber region [12], including switching adenoviral serotypes [13] or introducing therapeutic genes or immune stimulators in the adenoviral genome [14,15,16]. Moreover, combinations of oncolytic viruses with conventional therapies and/or immunotherapies have been investigated in an effort to enhance antitumor immune effects [17].

In the present study, we developed a novel OAd encoding a chimeric fusion protein of SA-4-1BBL that functions as a potent immunomodulatory and immune checkpoint activator. Our results show that OAdSA-4-1BBL induced oncolytic cell death mediated by activation of apoptosis and autophagy in two mouse lung cancer cell lines in vitro, and potent anti-tumor immune effects in vivo associated with increased tumoral infiltration of CD8^+^ T cells, NK cells, and a higher frequency of DCs.

## 2. Materials and Methods

### 2.1. Cell Lines and Culture Conditions

For the experimental study, we employed two murine respiratory cancer cell lines: TC-1 (Cat. CRL-2785, discontinued, ATCC, Manassas, VA, USA), derived from primary lung epithelial cells, and CMT64 (Cat. 10032301-1VL, Milipore-Sigma, Burlington, MA, USA), a primary alveogenic lung carcinoma cell line. As a control, we used the non-cancerous murine lung bronchial MM14.Lu cell line (Cat. CRL-6382, ATCC). For rOAd replication, we used the human embryonic kidney cell line HEK-293 (Cat. CRL-1573, ATCC). For the rOAds expansion, the human embryonic kidney cell line HEK-293 (Cat. CRL-1573, ATCC) was cultured and infected to obtain higher concentrations of virus to perform all the in vitro experiments.

The MM14.Lu, HEK-293, and CMT-64 cells were grown in DMEM medium (Cat. 10-013-CV, Corning Cellgro, Glendale, AZ, USA). TC-1 cells were cultured in RPMI-1640 medium (Cat. 10-040-CV, Corning). All media were supplemented with 10% fetal bovine serum and penicillin/streptomycin (10,000 IU/mL) (30-002-Cl, Corning, Glendale, AZ, USA) and maintained at 37 °C in 5% CO_2_.

### 2.2. Recombinant Adenoviral Vectors

The conditionally replicating adenoviruses (Ad5) were constructed using the AdenoQuick 2.0 kit from O.D.260 Inc. (Boise, ID, USA) containing a 24 bp deletion (Δ24) in the retinoblastoma protein (pRB), preventing replication in normal cells but not in cancer cells, which are characterized by inactive pRB (i.e., via constitutive phosphorylation or deletion) or by abnormal pRB control [18,19,20]. The expression cassette contains the cytomegalovirus (CMV) promoter driving the expression of SA or murine SA-4-1BBL as described previously [21]. These vectors are designated as OAdSA or OAdSA-4-1BBL, respectively. A replication-deficient adenoviral vector expressing green fluorescent protein (AdGFP) under regulation of the CMV promoter was used as a negative control for virus replication as described previously.

### 2.3. SA-4-1BBL Splenocyte Co-Stimulation In Vitro Assay

Splenocytes from C57BL/6 naïve mice were cultured (2 × 10^5^ cells/well) in 96-well U-bottom plates. These splenocytes were stimulated with an agonistic CD3 antibody at a suboptimal dose (0.25 µg/mL). The cultured cells were treated with increasing doses of protein (0.025–1.6 µL) concentrated from supernatant of TC-1 cells infected with AdGFP, OAdSA, or OAdSA-4-1BBL. The purified recombinant SA-4-1BBL protein was used as a positive control. The cells were incubated at room temperature for 1 h after treatment in the presence of naïve or SA-positive serum. After 48 h of culture, a pulse of [3H]-thymidine was added and incubated for an additional 16 h. After this time, the cells were harvested, and proliferation was evaluated as counts per minute from the DNA-associated reactivity using a beta plate counter as previously reported [22].

### 2.4. Evaluation of OAd-Mediated Oncolytic Cell Death

A total of 1 × 10^4^ cells were plated in 24-well plates and after 24 h infected at increased multiplicity of infection (MOI) concentrations. Adenoviral-mediated oncolytic cell death was evaluated at 72 h post-infection using the alamarBlue assay (Cat. DAL1025, Invitrogen, Waltham, MA, USA). After 1 h of incubation with alamarBlue, fluorescence in the samples was measured at 560/590 nm (excitation/emission) using a Synergy HT Multi-Mode Microplate Reader (Bio-Tek, Winooski, VT, USA). The fluorescence intensity (FI) values of each treatment were then normalized to mock (untreated) cells. The cytopathic effect (CPE) was also evaluated by bright-field microscopy. The images were taken at 20× magnification with the EVOS FL Imaging System (Thermo Fisher Scientific, Waltham, MA, USA).

### 2.5. Western Blot Analysis

After treatment, the cells were harvested to recover whole lysate protein using RIPA buffer. After obtaining cell lysate, the protein was centrifuged, and its concentration was determined using the PIERCE BCA Protein Assay kit (Cat. 23225, Thermo Fisher Scientific). The protein samples were heated and loaded in a 10% or 12% SDS-polyacrylamide gel and run in an electrophoresis chamber. Protein from the gels was transferred to PVDF membranes (Cat. 10600023, GE Healthcare Life Sciences, Pittsburgh, PA, USA). The following primary antibodies were used to detect the desired proteins for each membrane: rabbit anti-streptavidin (SA) polyclonal antibody (Cat. ab6676, Abcam, Cambridge, MA, USA), mouse anti-adenovirus type 5 E1A monoclonal antibody (Cat. 554155, BD Pharmigen, San Diego, CA, USA), goat anti-4-1BB Ligand/TNFSF9 polyclonal antibody (Cat. AF1246, R&D Systems, Minneapolis, MN, USA), rabbit anti-LC3 polyclonal antibody (Cat. L7543, Millipore Sigma), mouse anti-SQSTM1/p62 monoclonal antibody (Cat. ab56416, Abcam), mouse anti-caspase-3 (Cat. 9662, Cell Signaling, Danvers, MA, USA), mouse anti-cleaved caspase-3 (Cat. 9661, Cell Signaling), and rabbit anti-actin (Cat. A2066, Millipore Sigma). Next, the membranes were incubated with anti-mouse immunoglobulin G (IgG) (Cat. 31430, Thermo Fisher Scientific), anti-rabbit IgG (Cat. 31460, Thermo Fisher Scientific), or anti-goat IgG (Cat. 6160-05, Southern Biotech, Birmingham, AL, USA), horseradish peroxidase-linked, species-specific whole antibody. For luminescence production, the substrate ECL reagent was used (Cat. RPN3244, GE Healthcare Life Sciences). Quantification of the densitometry of the bands was performed using the image analysis software ImageJ v1.54d and the values for each group compared with one-way ANOVA statistical analysis (Prism 10, GraphPad Software Inc., La Jolla, CA, USA).

### 2.6. Quantification of Activated Autophagy-Mediated Adenosine Triphosphate (ATP) Release

The mouse lung cancer cells were infected with AdGFP, OAdSA, or OAdSA-4-1BBL at a MOI concentration of 25 for 48 h. For extracellular ATP detection, supernatants from virus-infected cells were collected 48 h after infection and extracellular ATP was measured with a bioluminescence assay ATP Detection Kit (Cat. A22066, Thermo Fisher Scientific). ATP release was measured at maximum emission of 560 nm.

### 2.7. Evaluation of OAds-Mediated Cytotoxicity and Virus Replication in Lung Non-Cancerous Cells

The TC-1, CMT64, or MM14.Lu cell lines were plated at a cell density of 5 × 10^5^/dish and after 24 h cells were not infected (mock) or infected with AdGFP, OAdSA, or OAdSA-4-1BBL at an MOI concentration of 50. At 24 h post-infection, Ad E1A protein expression was detected by Western blot assay. In a separate experiment, these cell lines were plated at a cell density of 1 × 10^4^/well in a 24-well plate and later infected as described above. At 72 h post-infection, cell viability was evaluated using the alamarBlue assay. Simultaneously, the same culture conditions were used to quantify the viral replication using the Adeno-X™ qPCR Titration Kit (Cat. 632252, Takara, San Jose, CA, USA).

### 2.8. Oncolytic Ads Anti-Tumor Therapeutic Efficacy Study

C57BL/6 female mice (6–8 weeks old) were purchased from The Jackson Laboratory (Bar Harbor, ME, USA). The animals were housed in the animal facility of the University of Missouri following the current guidelines established by the NIH (Guide for the Care and Use of Laboratory Animals, NIH Publication No. 8023, Rev. 1978).

The animals were subcutaneously inoculated in the flank with 1 × 10^5^ TC-1 cells in a total volume of 100 µL of PBS. After nine days, tumors were palpable and the animals were randomly assigned to the treatment groups: PBS, OAdSA, or OAdSA-4-1BBL. Once the tumor was established, a treatment scheme of three intratumoral injections was administered every three days at a dose of 1 × 10^9^ ifu/50 µL (infective units). The tumor size was measured every three days with a caliper in two dimensions (length and width) for a maximum of 26 days after inoculation. Tumor volume was estimated by the following equation: V = (L × W2)/2, where V is volume, L is length, and W is width.

### 2.9. Immunohistochemistry

The tumors were collected from the animal groups and fixed in 10% formalin, embedded in paraffin blocks, and processed for hematoxylin and eosin (H and E) and immunohistochemistry (IHC) staining. For IHC, slides were incubated with mouse anti-adenovirus E1A (1:200) (Cat. Ab204123, Abcam) or mouse anti-SA IgG antibodies. To detect the presence of the target protein, we incubated with the ultra-sensitive ABC peroxidase standard kit (Cat. 32050, Pierce, Rockford, IL, USA). The same tissue slides were counter stained with H and E. Images were obtained using a Leica Aperio ImageScope at 40× magnification (Leica Biosystems, Buffalo Grove, IL, USA).

### 2.10. Flow Cytometry and Phenotyping

We followed the same treatment scheme for the OAd therapeutic efficacy study. Five days after the last virus injection, we collected the tumoral infiltrating cells for analysis from at least 8 animals per group. Tumor tissues were collected in 10% FBS DMEM, weighed, and minced for incubation with 1 mg/mL collagenase (Cat. C5138, Milipore-Sigma) and 200 U DNase I (Cat. D5025, Milipore-Sigma) at 37 °C for 30 min to obtain single cells and release the infiltrating tumor cells. Then, the single cells were blocked with anti-mouse CD16/32 Fc Block 1:50 ratio (Cat. 553142, BD Pharmingen) for 20 min. the single-cell suspensions were stained with fluorescent-conjugated antibodies to various cell surface markers (Appendix A). Sample acquisition was performed using an Aurora Cytek system (Fremont, CA, USA). Cell percentages and absolute numbers were calculated and reported as in [5].

### 2.11. Statistical Analysis

All data sets were analyzed using one- or two-way ANOVA considering a normal (Gaussian) distribution followed by a Tukey test for multiple comparison to identify specific differences between treatment groups. The statistical analysis was performed using Prism software v10.0.3 (GraphPad Software Inc.).

## 3. Results

### 3.1. Recombinant Oncolytic Adenovirus Expresses an Active form of SA-4-1BBL

The recombinant oncolytic adenoviruses (rOAds) used in this study are shown in Figure 2A. SA-4-1BBL is a novel version of 4-1BBL that produces soluble oligomers and binds to the 4-1BB (CD137) receptor. Activation by this protein has pleiotropic effects on cells of innate, adaptive, and regulatory immunity [3]. We first tested the capacity of rOAds to produce the encoded proteins, SA and SA-4-1BBL, extracellularly, as well as their ability to form oligomers in vitro. For this purpose, TC-1 cells were infected with OAdSA or OAdSA-4-1BBL at a MOI of 25. At 24 h post-infection, SA or SA-4-1BBL expression was evaluated by Western blot. SA-4-1BBL forms oligomers that can be dissociated into monomers via denaturation by heat [3]. An immune blot using an antibody against SA showed a ~15 kDa band corresponding to SA in the OAdSA lane. Similarly, a ~37 kDa band corresponding to SA-4-1BBL monomers was detected in samples that were additionally denatured by heat. SA-4-1BBL monomers were observed at the same molecular weight as the recombinant SA-4-1BBL protein that was used as a positive control. Interestingly, SA-4-1BBL oligomers were detected in non-heated samples and matched the expected molecular weight of ~120 kDa, similar to the oligomeric form of the recombinant SA-4-1BBL protein (Figure 2B). For the SA protein with a monomeric size of 17 kDa, we observed a faint band in the non-heated sample of ~37 kDa, suggesting this control protein can associate to form oligomers.

Next, we evaluated the ability of SA-4-1BBL released by the infected cells to stimulate splenocyte proliferation in vitro. TC-1 cells were infected with a replication-defective AdGFP, as a negative control, and OAdSA or OAdSA-4-1BBL at a MOI of 50. At 72 h post-infection, OAd-infected TC-1 cells displayed a complete cytopathic effect (CPE), whereas no CPE was observed in AdGFP-infected cells (Figure 2C). The supernatants from each treatment were collected by centrifugation, filtration, and concentrated by centricon centrifugation. Equal amounts of protein were then added to 2 × 10^5^/well splenocytes from healthy naïve C57BL/6 mice and incubated in the presence of a suboptimal dose of anti-CD3 antibody, which provides the direct T-cell receptor union to allow SA-4-1BBL-induced co-stimulation. The OAd-produced protein, SA-4-1BBL, showed significant stimulatory activity, resulting in a two-fold increase in splenocyte proliferation compared to controls, GFP and SA (Figure 2D). The recombinant SA-4-1BBL, used as a positive control, showed the highest splenocyte proliferation by three-fold compared to controls. Altogether, these results indicate that OAdSA-4-1BBL efficiently expresses SA-4-1BBL, forming soluble oligomers with biological activity as a costimulatory protein.

### 3.2. OAdSA and OAdSA-4-1BBL Induce Cytotoxic Effect on Lung Cancer Cell Lines In Vitro

We tested two murine lung cancer cells, TC-1 and CMT-64, which have been demonstrated to support OAds infection and replication [23,24]. The cells were non-infected (mock) or infected with AdGFP, OAdSA, or OAdSA-4-1BBL at increasing MOI concentrations. At 72 h post-infection, the alamarBlue assay revealed that rOAds induced cytotoxicity in a dose-dependent manner. The MOI of 25 in OAdSA and OAdSA-4-1BBL decreased cell viability by ~50% in TC-1 and ~60% in CMT64 cell lines. The lowest viability levels were observed using a MOI of 200 with measurements below 10% for both cell lines (Figure 3A).

In the same treatment groups, we evaluated the expression of the Ad E1A (35–46 kDa), a key component of Ad replication machinery [25], as well as the expression of SA-4-1BBL (37 kDa) in both lung cancer cell lines. At 48 h post-infection, an immunoblot revealed that only cells infected with OAdSA or OAdSA-4-1BBL showed Ad E1A expression, and using an antibody against murine 4-1BBL, we detected strong SA-4-1BBL production in cells infected with OAdSA-4-1BBL (Figure 3B). These results indicate that our novel OAds can efficiently replicate, express, and deliver either SA or SA-4-1BBL in mouse lung cancer cells.

### 3.3. Activation of Apoptotic Cell Death Is Associated with rOAds-Mediated Oncolytic Cell Death

As we observed a potent killing effect on lung cancer cells produced by the rOAds infection, we investigated the involvement of type I programmed cell death (apoptosis) by measuring activated cleaved caspase-3 (Cas3) expression. Both TC-1 and CMT64 cells were infected as described above at a MOI concentration of 100 for 48 h. An immunoblot showed that all groups expressed the full-length caspase-3 (35 kDa). In contrast, only cells infected with rOAds expressed high levels of the cleaved Caspase-3 protein (17 kDa), which is indicative of apoptosis activation (Figure 3C). These bands were analyzed by image densitometry, which revealed a trend to increase by more than 1.5-fold the quantification of cleaved Caspase-3 on TC-1 in both rOAds groups. This increase is even higher on CMT64 cells showing a ~20-fold increase when compared to the mock control (Figure 3D). These results suggest that the rOAd-mediated killing effect is associated with apoptosis activation.

### 3.4. rOAds Trigger Autophagy-Mediated Adenosine Triphosphate (ATP) Release in Lung Cancer Cell Lines In Vitro

It is widely documented that OAds induce autophagy, a type II programmed cell death, in different cancer cell lines [26]. Therefore, to better understand the involved pathways leading to cell death after the infection with these new constructs, OAdSA or OAdSA-4-1BBL, we evaluated the capability of these viruses to induce autophagy in murine lung cancer cells.

During the process of autophagy, free soluble LC3-I is cleaved to its active form LC3-II by the Cys-protease autophagin (Atg-4). After this conversion, LC3-II binds the membrane and fuses with more LC3-II proteins to form an autophagic vacuole [27]. We evaluated the induction of autophagy after rOAds infection by detecting the conversion from LC3-I to LC3-II in a Western blot using whole cell lysate protein. We observed the presence of a band corresponding to LC3-I (19 kDa) and a lower band for LC3-II (17 kDa). Both cell lines, TC-1 and CMT64, showed more LC3-II protein than mock and AdGFP controls, indicating a higher conversion from LC3-I (Figure 4A). To further validate autophagy induction, p62/sequestosome-1 (SQSTM1) protein expression was also evaluated. p62/SQSTM1 is a ubiquitin-binding scaffold protein that interacts with ubiquitinated protein aggregates, binds to LC3 and the GABA type A receptor-associated protein (GABARAP), and is degraded after fusion of autophagosomes with lysosomes [28]. Thus, declined expression of p62/SQSTM1 positively correlates with autophagic flux. An immunoblot analysis revealed a 62 kDa band corresponding to p62/SQSTM1. Interestingly, a lower amount of this protein was observed in TC-1 cells infected with OAdSA or OAdSA-4-1BBL when compared to mock- and AdGFP-infected cells (Figure 4A). However, decreased p62 levels were not detected for CMT64 cells. The densitometry quantification of these bands confirmed a higher conversion rate of LC3 from I to II in both cell lines treated with OAdSA-4-1BBL, correlating with a marked decrease of p62/SQSTM1 on TC-1 (Figure 4B). These results suggest that there is a component of autophagy-mediated cell death in the resulting oncolysis after infection of both OAds constructs.

A late indicator of autophagy is ATP secretion to the extracellular media, which is a marker of DAMPs [29,30]. Oncolytic viruses induce ICD, which may provide potent and long-lasting anticancer immunity [9]. Therefore, we evaluated the effect of OAdSA and OAdSA-4-1BBL to induce the release of ATP, an ICD marker. Lung cancer cell lines were non-infected (mock) or infected with AdGFP, OAdSA, or OAdSA-4-1BBL at a MOI of 50. At 72 h following infection, the release of extracellular ATP was measured using a luminescence assay. TC-1 cells infected with OAdSA and OAdSA-4-1BBL showed a ~seven-fold and ~six-fold increase in extracellular ATP, respectively, when compared to the mock control. Similar findings were observed in the CMT64 cells showing increased ATP release by ~4.5-fold for OAdSA and ~3.5-fold for OAdSA-4-1BBL, compared to the mock group. AdGFP infection did not increase ATP release (Figure 4C). This result suggests that rOAd-induced autophagy leads to ATP release.

### 3.5. rOAds Preferentially Infect, Replicate, and Kill Murine Lung Cancer Cell Lines, but Not Non-Cancerous Lung Cells

To evaluate whether our novel rOAds have cancer selectivity, murine lung cancer and non-cancerous cells were infected with AdGFP, OAdSA, or SA-4-1BBL at an MOI of 50 to produce a moderate effect—as we observed in the previous section. After 72 h of infection, viability was quantified via the alamarBlue assay, revealing that OAdSA and OAdSA-4-1BBL induced CPE in both TC-1 and CMT64 cells, whereas no or slight CPE was observed in non-cancerous MM14.Lu cells. In TC-1 cells treated with OAdSA or OAdSA-4-1BBL, the percentage of cell viability was 51% and 55%, respectively. A similar effect was observed in CMT64 cells, where treatment with OAdSA or OAdSA-4-1BBL resulted in 34% and 42% of cell viability, respectively. In contrast, non-cancerous cells (MM14.Lu) were mostly unaffected by OAdSA or OAdSA-4-1BBL infection, conserving 86% and 91% of cell viability, respectively (Figure 5A). As shown in previous results, AdGFP did not induce CPE.

The cell death resistance of the non-cancerous MM14. Lu cells can be explained by a decreased virus replication, as OAds are designed to use oncogenes as replication promoters that are upregulated only in cancer cells. To evaluate the viral replication of each cell line, we quantified the number of viral copies in the supernatant after 24 h of infection using qPCR. In the non-cancerous MM14. Lu cell line, OAds were quantified at ~2000 copies per µL, compared to ~200 copies/µL for AdGFP, with no significant difference. However, in both cancer cell lines, TC-1 and CMT64, OAdSA and OAdSA-4-1BBL replication was greatly increased, more than 10-fold higher than the viral copies produced in the non-cancerous cell line. CMT64 was the cell line with the higher number of viral copies, indicating greater sensitivity to OAd infection (Figure 5B).

This increased viral replication observed only in the cancer cell lines was additionally confirmed by an immunoblot assay targeting the Ad E1A proteins. The Ad E1A region encodes two closely related gene products: 243 and 289 amino acid phosphoproteins. These proteins differ in their primary sequence only by 46 amino acids, which are unique to the 289 amino acid protein [31]. In TC-1 and CMT64 cells infected with OAdSA or OAdSA-4-1BBL, strong bands were detected for both E1A subunits while band intensity for MM14.Lu cells was significantly decreased. As expected, no bands were observed in cells infected with AdGFP (Figure 5C). These findings correlate with high OAd replication observed from qPCR analysis on the cancer cell lines, without affecting the integrity of the non-cancerous cell line.

### 3.6. OAdSA-4-1BBL Efficiently Delivers SA-4-1BBL, Replicates, and Has a Potent Therapeutic Efficacy against Established TC-1 Tumors

Figure 6A depicts the therapeutic regimen used in this pre-clinical tumor model. TC-1 tumor inoculation and OAd treatment formed two groups, and different ending timepoints were chosen. The first timepoint was on day 18 to use the tumor for IHC, and the second one was on day 24 to compare the tumor sizes between the different treatment groups.

To determine if OAdSA or OAdSA-4-1BBL can efficiently deliver and express their respective transgene and sustain productive viral replication, TC-1 tumor-bearing mice were injected intratumorally (i.t.) with PBS, OAdSA, or OAdSA-4-1BBL at 1 × 10^9^ ifu. The IHC staining revealed high expression of streptavidin, and Ad E1A was only present in the tumors of mice treated with OAdSA or OAdSA-4-1BBL. In contrast, expression of streptavidin or Ad E1A was not detected within tumors from PBS-treated mice (Figure 6B). This finding indicates that both OAdSA and OAdSA-4-1BBL can efficiently infect, replicate, and propagate within the tumor mass while producing their encoded protein. To evaluate whether OAdSA-4-1BBL has an increased advantage over OAdSA in an immunocompetent host to stop or eradicate the growth of an established TC-1 tumor, mice received a set of i.t. injections when tumors became palpable, starting at day 9 after inoculation. Injections were administered for PBS, OAdSA, or OAdSA-4-1BBL at a concentration of 1 × 10^9^ ifu. Tumor growth was followed every 3 days, and a statistically significant difference in tumor volume was observed at day 15 between OAdSA-4-1BBL- and PBS-treated mice. At the end of the experiment on day 24, tumor size reduction was approximately 50% in OAdSA-treated mice, whereas in OAdSA-4-1BBL, it was ~78% compared to PBS-treated mice (Figure 6C; *p* < 0.001). This increased antitumoral effect by OAdSA-4-1BBL suggests a co-adjuvant effect by SA-4-1BBL in addition to the oncolytic activity of the virus.

### 3.7. Immune Cell Phenotyping of Tumors Demonstrates the Activation of the Innate and Adaptive Immune Response by OAd Expressing the Costimulatory Protein SA-4-1BBL

To determine whether the increased OAdSA-4-1BBL anti-tumor therapeutic efficacy was associated with higher immune system activation, we performed tumor-infiltrated immune cell phenotyping via flow cytometry analysis. The mice were assigned to treatment groups receiving PBS, OAdSA, or OAdSA-4-1BBL. Five days after the last virus injection, tumors were recovered and processed to obtain a single cell suspension, which was later stained to detect distinct immune cell populations. From the CD45^+^ population, t-distributed stochastic neighbor embedding (t-SNE) plots were generated for each of the major immune cell population markers.

The t-SNE comparison revealed the highest density of CD8^+^ cells, natural killer cells, macrophages, and dendritic cells in the OAdSA-4-1BBL-treated tumors, followed by OAdSA treatment and then PBS treatment (Figure 7A). Below each t-SNE plot, we show the quantification for the respective cell population for each group (Figure 7B). The tumor infiltration percentage of CD8^+^ T cells and CD11c^+^ dendritic cells is significantly higher in the OAdSA-4-1BBL-treated group compared to the OAdSA group, suggesting higher cytotoxic immune system recruitment mediated by the production of SA-4-1BBL. Moreover, when the OAdSA-4-1BBL group is compared to the PBS group, we observed a significant decrease in the percentage of CD19^+^ B cells, a lower CD4^+^/CD8^+^ ratio, an increased percentage of NK1.1^+^ natural killer cells, F4/80^+^ macrophages, as well as a higher percentage of CD3^+^ lymphocytes from the CD45-positive population. Collectively, these results strongly indicated that the produced SA-4-1BBL protein triggered immune system activation, resulting in increased immune cell infiltration into the tumors. This result also suggests that OAdSA-4-1BBL significantly reprograms the tumor microenvironment compared to PBS-injected tumors.

## 4. Discussion

Co-stimulation of the 4-1BB (CD137) receptor induces high levels of activation of CD4^+^ and CD8^+^ T cells, resulting in an enhanced antitumoral response [22]. In recent years, investigators developed multiple approaches to activate the 4-1BBL/4-1BB costimulatory pathway. One strategy is the use of agonistic antibodies to 4-1BB (CD137) [32,33], which have shown therapeutic efficacy as first-generation agonistic antibodies to 4-1BB alone or in combination with other anticancer agents in various preclinical studies [1,34,35]. However, the major drawback of this strategy against 4-1BB is the high level of systemic toxicity observed in preclinical and clinical studies [22,33]. However, a recently developed antibody lacking the Fc region demonstrated high-avidity binding to 4-1BB, and potent costimulatory capacity without inducing systemic inflammatory cytokine production or hepatotoxicity associated with IgG-based 4-1BB agonists [36].

On the other hand, the use of oncolytic viruses as a cancer therapy has gained increased interest in the latest years. Adenoviruses are the most studied and the first to be used in a clinical trial since 2005 [37]. OAds can selectively replicate, spread, and destroy tumor cells [6]. Oad therapies have the advantage of being immunogenic, stimulating APCs, and amplifying the direct tumor lysis by further recruiting immune cells by secreting TAAs and DAMPs, which activate ICD [38].

To take advantage of this immune stimulatory mechanism, several studies have focused on developing different versions or modalities of viral vectors to deliver 4-1BBL [39,40,41,42]. These studies, although interesting, have some limitations, such as modest tumor regression [39]. Three studies used the mouse melanoma cell line B16-F10 [39,40,41], and one study used a pancreatic xenograft model [42], which lacks a fully active immune system. Another study used a replication-defective Ad expressing the membrane form of 4-1BBL. Even though these 4-1BBL oligomers can remain active after the cell’s death by being released as apoptotic vesicles, these vesicles are primarily marked for degradation and are mostly phagocytized by surrounding macrophages, thus limiting their effect on tumor destruction and cancer cell eradication [41]. To overcome the limitations of using the transmembrane form of 4-1BBL that depends on cell contact activation, we developed of a chimeric active costimulatory fusion protein by linking core streptavidin to the extracellular functional domain of 4-1BBL (SA-4-1BBL) [4,43]. This molecule can be released in the tumor environment where it forms oligomers, and in soluble and active form it has pleiotropic effects on cells of innate, adaptive, and regulatory immunity [4,43].

Recently, an OAd expressing SA-4-1BBL fused to human papilloma virus E7 that also includes a signal peptide to target the fusion protein into the SA-4-1BBL fused to E7 displayed greater anti-tumor efficacy than a DNA vaccine expressing the same fusion protein [44]. However, this report did not describe OAd-induced cell death mechanisms or tumor infiltrating immune populations to relate the observed antitumor effect with the activation induced by SA-4-1BBL.

We highlight the importance of evaluating the OAdSA-4-1BBL therapeutic efficacy in an immunocompetent syngeneic mouse cancer model, as an intact immune system is necessary to enhance the costimulatory properties of SA-4-1BBL. By using OAd as an expression vector, we are abolishing the time- and cost-intensiveness of SA-4-1BBL production, isolation, and purification [21].

In this report, we describe a novel strategy for cancer immunotherapy using an OAd armed with the immune checkpoint stimulator SA-4-1BBL. This strategy takes advantage of the robust immunomodulatory features of SA-4-1BBL [45]. We show that OAdSA-4-1BBL produces an active soluble oligomer capable of stimulating splenocytes in vitro. Infection with this OAd in the murine lung cancer cells, TC-1 and CMT64, induces oncolytic cell death by the activation of autophagy detected by increased LC3-II conversion, decreased p62, caspase-3 mediated apoptosis, and ATP release via DAMPs signaling. We showed our rOAds can efficiently infect, kill, and replicate in the murine cancer cell lines as demonstrated by the production of E1A protein and the detection of higher number of OAds copies in the cancer cell lines by the qPCR quantification. This capacity of infection and replication of human OAds in murine cell lines has been previously supported by other groups, which can be related to the expression of other viral proteins that support DNA replication like HPV E6 or E7 for TC-1 cells [23,24]. Similar findings were reported using an oncolytic respiratory syncytial virus [46]. Hence, it is important to report these mechanisms using our novel rOAds on lung cancer cells.

Most importantly, OAdSA-4-1BBL exhibited potent anticancer effects in a murine TC-1 subcutaneous tumor model. This effect was associated with the oncolytic effect of the OAds and immune system stimulation by the chimeric protein SA-4-1BBL, which induced vital recruitment of immune cell populations into the tumor. There was also a higher presence of APC dendritic cells and macrophages, and cytotoxic mediated antitumor activity by CD8^+^ T cells and natural killer cells. This higher immune cell infiltration corresponds with the costimulatory signature of the 4-1BB/4-1BBL pathway [4,47,48]. This increase in cytotoxic activity is also shown in Appendix A, where the proportion of all the analyzed immune cell’s populations are detailed from the total tumoral infiltrating immune cells (CD45^+^). It would be interesting for a future study to evaluate if localized expression of SA-4-1BBL in infected tumoral cells can have a systemic effect, by the detection of activated cells in draining lymph nodes or splenocytes.

As a result of the promising findings of this work, we are interested in evaluating the possibility of further enhancing the antitumoral efficacy of OAdSA-4-1BBL by combining it with immune checkpoint inhibitors, chemotherapy, or drugs that can enhance OAd efficacy. We will explore the potential benefits of using OAds to treat different cancers, like melanoma, breast cancer, or primary and metastatic lung cancer. Additionally, we consider the application in a clinical setting of this treatment by delivering OAds to treat lung cancer using formulations administered via inhalers or nebulizers. These approaches could present low-cost alternatives to deliver vectors for the expression of recombinant proteins that function as a coadjuvant cancer therapy.

## Figures and Tables

**Figure 1 vaccines-12-00340-f001:**
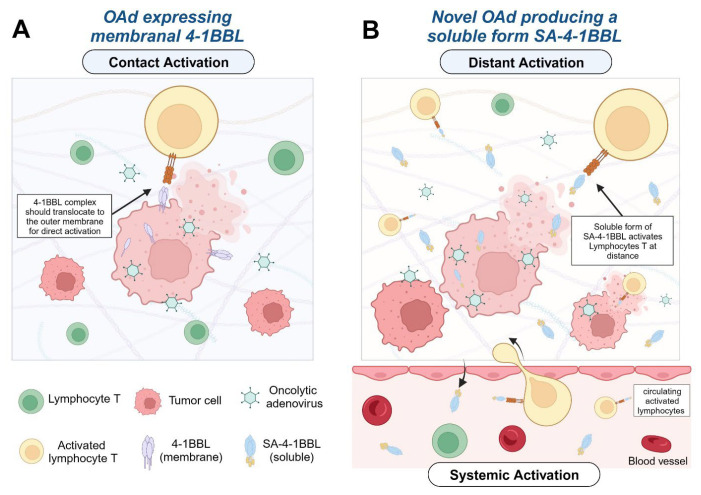
Immune cell activation by the production and release of SA-4-1BBL via oncolytic cell death of infected tumoral cells. (**A**) Traditional cell-to-cell contact mediated immune activation of the 4-1BB/4-1BBL pathway. (**B**) Our chimeric molecule, SA-4-1BBL, is produced inside the infected tumoral cells and released during oncolysis, activating the immune system at distance via the active soluble form of the protein. Created with Biorender.

**Figure 2 vaccines-12-00340-f002:**
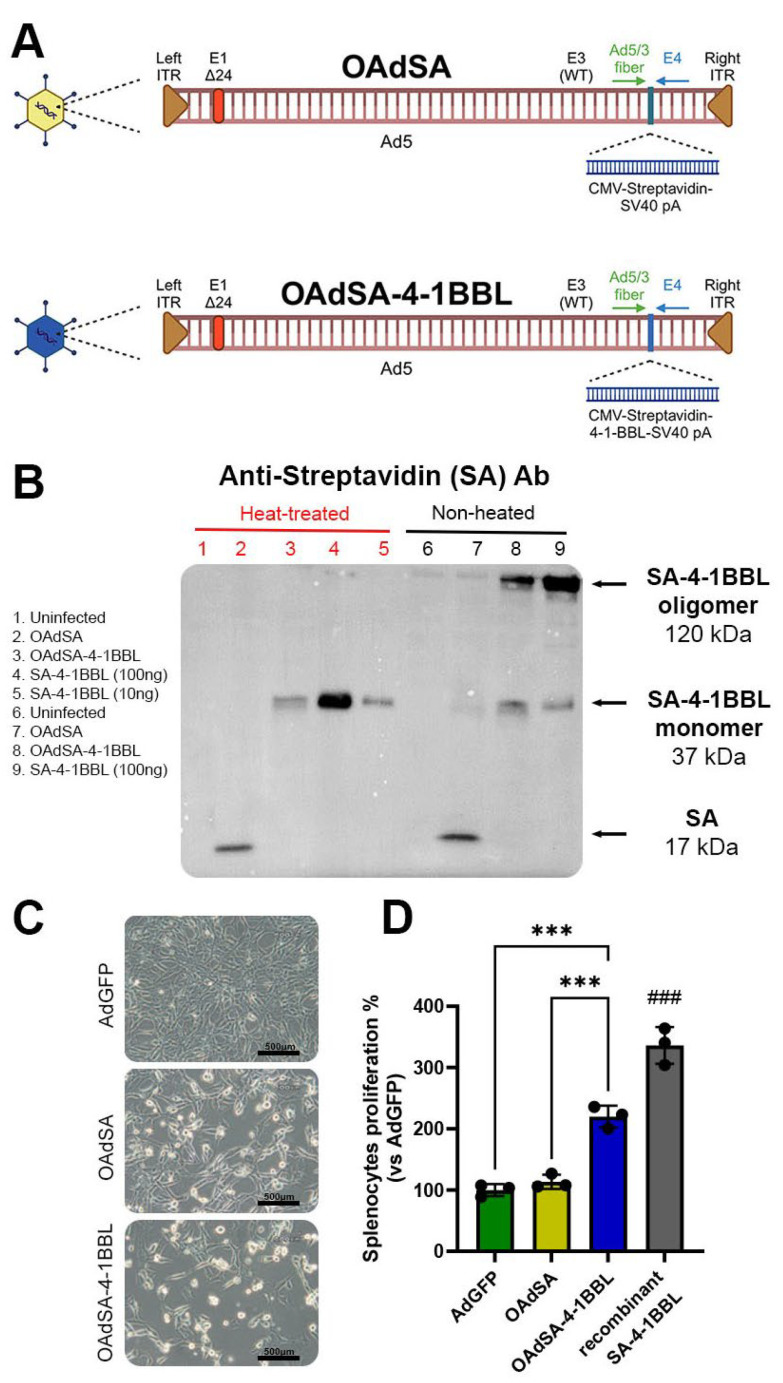
Recombinant OAdSA-4-1BBL efficiently produces an active form of SA-4-1BBL capable of stimulating splenocyte proliferation. (**A**) Scheme of the recombinant OAds constructions encoding SA and SA-4-1BBL. Created with Biorender. (**B**) Immunoblot against SA showing the production of SA and SA-4-1BBL proteins by infected TC-1 cells, and the formation of oligomers in non-heated samples. (**C**) Cytopathic effect over TC-1 after OAd infection. (**D**) The supernatant obtained from panel C was used to stimulate the proliferation of murine splenocytes. (Graph shows mean ± SD, black dots represent an individual experiment’s measure, *** *p* < 0.001, ^###^ *p* < 0.001 vs. all other groups).

**Figure 3 vaccines-12-00340-f003:**
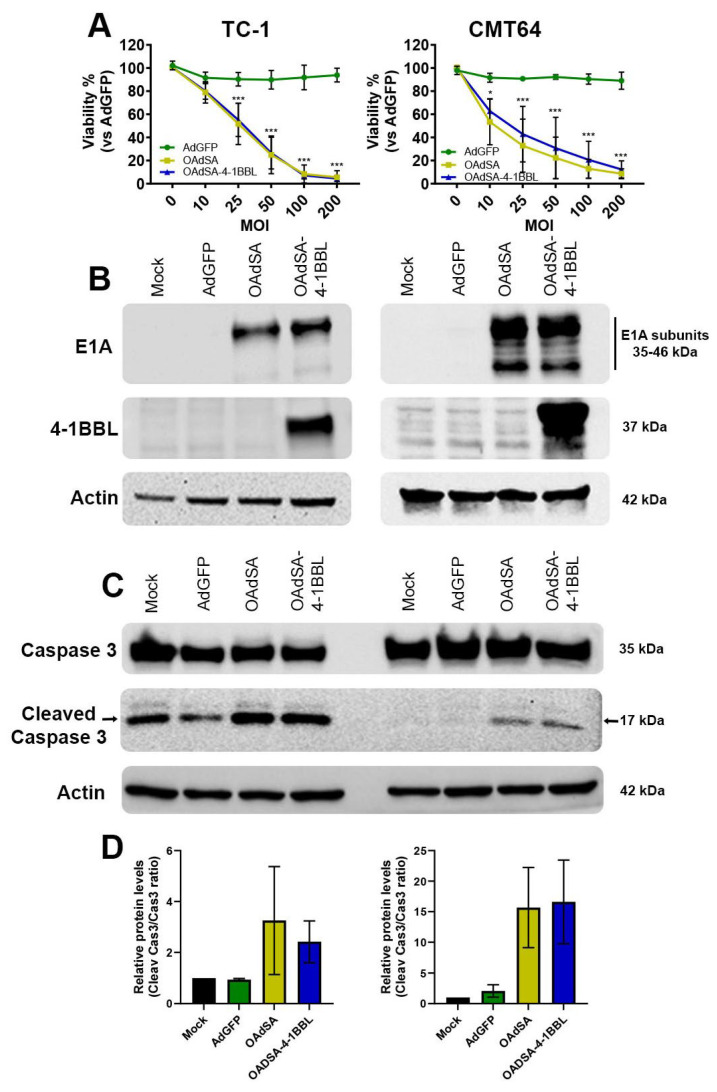
rOAds infection leads to decreased lung cancer cell viability, with active viral replication and activation of apoptotic cell death. (**A**) rOAds-mediated killing effect was evaluated using the alamarBlue cell viability assay. (**B**) Ad E1A and SA-4-1BBL expressions were assessed by Western blot at 48 h post-virus infection. (**C**) Expressions of full-length and cleaved caspase-3 were evaluated by Western blot at 48 h post rOAd infection. (**D**) Cleaved caspase-3 activation on rOAd groups was quantified by an immunoblot band. (Graphs shows mean ± SD, * *p* < 0.05, *** *p* < 0.001).

**Figure 4 vaccines-12-00340-f004:**
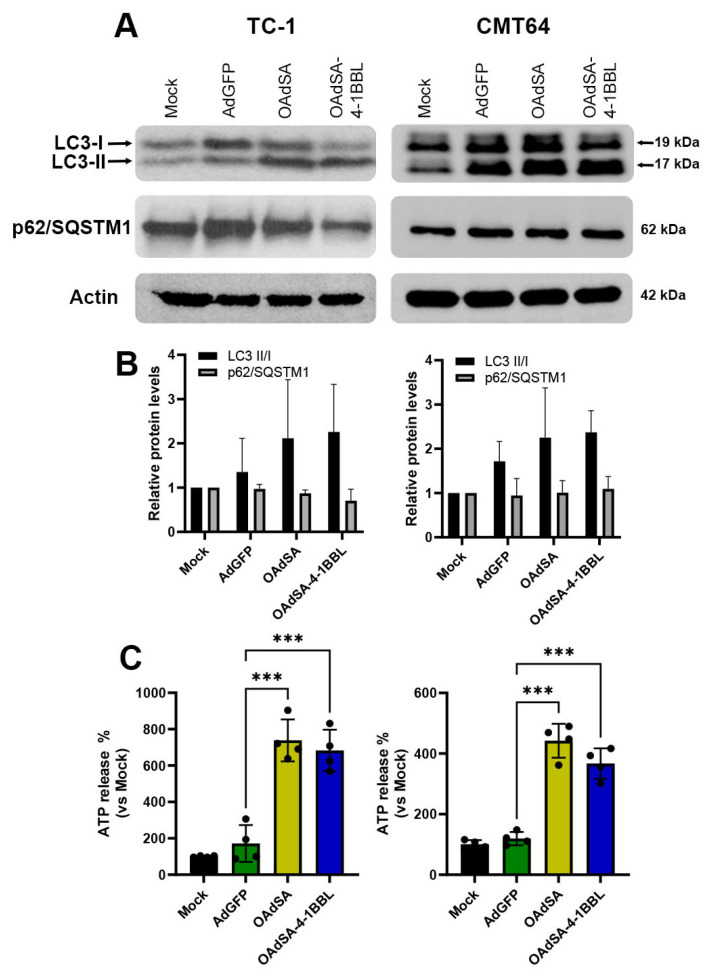
rOAds trigger autophagy activation and ATP release in lung cancer cells. (**A**) Immunoblots showing the conversion of autophagy marker LC3-I to LC3-II at 48 h post-rOAd infection in TC-1 and CMT64 murine lung cancer cell lines. The p62 protein was decreased in TC-1 cells, characteristic of autophagosome degradation. (**B**) Densitometry quantifications of the autophagy-associated protein bands. Quantified by an immunoblot band. Actin was used as a loading control. (**C**) ATP release was assessed 72 h after rOAd infection. (Graphs show mean ± SD, black dots represent an individual experiment’s measure, *** *p* < 0.001).

**Figure 5 vaccines-12-00340-f005:**
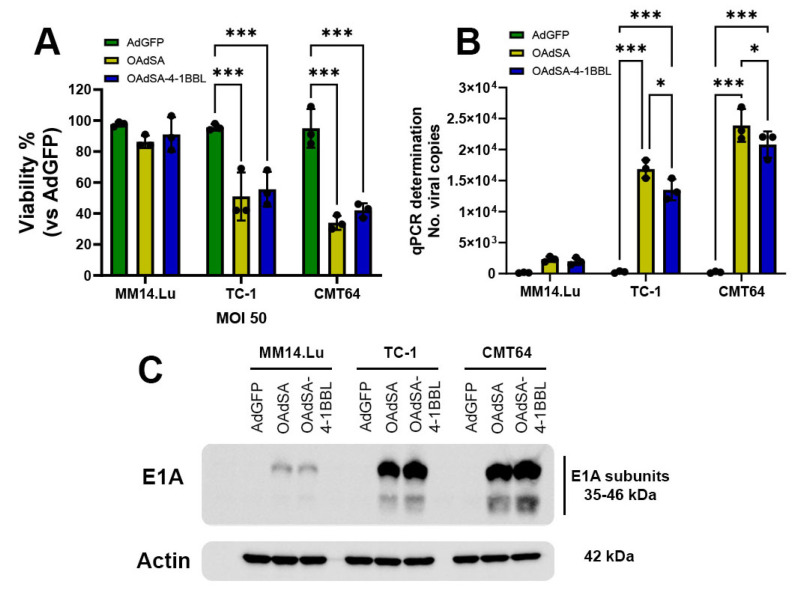
Evaluation of recombinant OAds cancer cell selectivity. (**A**) Murine lung cancer TC-1 and CMT64 and non-cancerous MM14. Lu cells were infected for 72 h with AdGFP, OAdSA, or OAdSA-4-1BBL to evaluate cytopathic effect. (**B**) After 24 h of infection, supernatants were used to determine adenovirus yield from each cell line by qPCR adenovirus genome detection. (**C**) Ad E1A protein expression was detected with an anti-adenovirus type 5 E1A monoclonal antibody. Actin was used as a loading control. (Graphs show mean ± SD, black dots represent an individual experiment’s measure, * *p* < 0.05, *** *p* < 0.001).

**Figure 6 vaccines-12-00340-f006:**
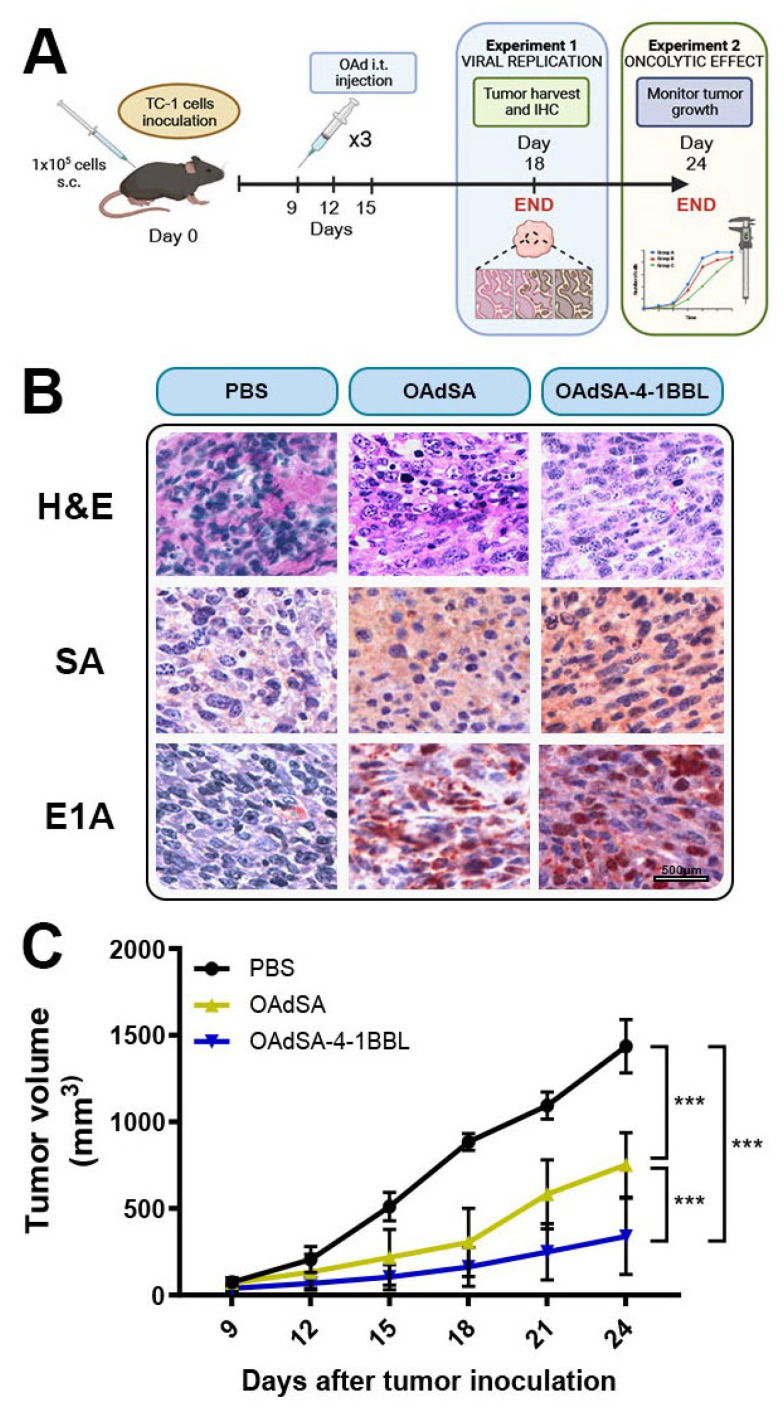
OAdSA-4-1BBL shows potent therapeutic efficacy against established TC-1 tumors. (**A**) In vivo treatment scheme. Created with Biorender. (**B**) Representative IHC staining of SA and Ad E1A from TC-1 tumor sections. Tumors were counterstained with H and E. (**C**) TC-1 tumor volume reduction after administering rOAds intratumorally in C57BL/6 mice. (Graph shows mean ± SD, *** *p* < 0.001).

**Figure 7 vaccines-12-00340-f007:**
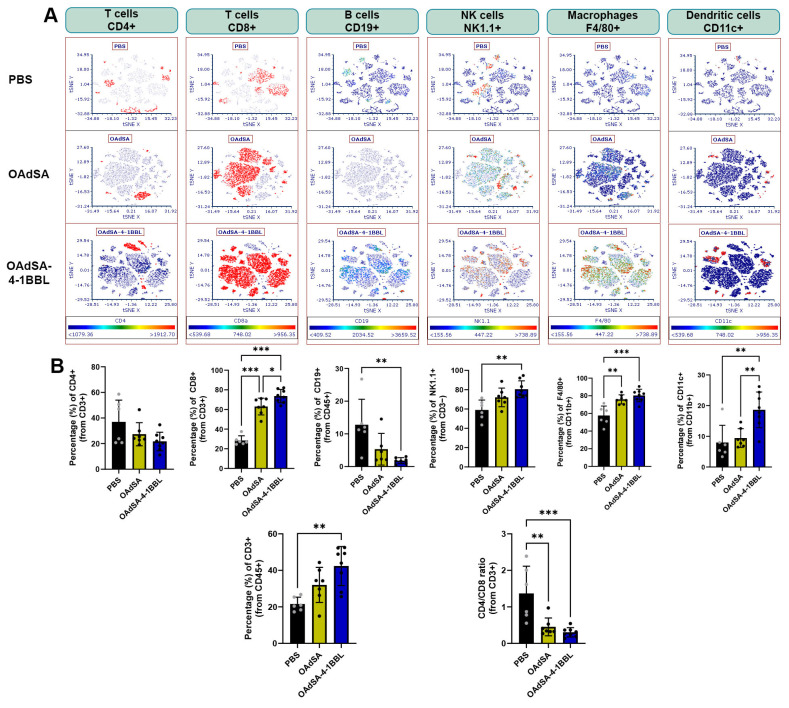
Immune cell phenotyping of tumor-infiltrating cells. (**A**) t-SNE plots comparing the presence of the main immune populations between treatment groups. (**B**) Quantification of flow cytometer’s infiltrating immune cell populations in the tumor. (Graphs show mean ± SD, black dots represent an individual sample’s measure Mean ± SD, * *p* < 0.05; ** *p* < 0.01; *** *p* < 0.001).

## Data Availability

No data set was generated in the present study. All current data are presented in the above figures. Experimental data are available upon reasonable request.

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
