# Peer review of "Oncolytic Adenovirus Armed with a Novel Agonist of the CD137 Immune Checkpoint Stimulator Suppresses Tumor Growth"

_vaccines, 2024, doi:10.3390/vaccines12030340_

Round 1

Reviewer 1 Report

Comments and Suggestions for Authors

The focus of this study is to demonstrate that novel oncolytic adenovirus with soluble agonist of CD137 is more efficacious and improves anti tumor effect and immune infiltration. The manuscript findings are supported by existing literature.

The findings in this manuscript are significant, contributing to the understanding of immune suppression in lung cancer and investigating a novel adenovirus as a means to overcome it which has demonstrated a robust anti-tumor activity in a lung cancer model.

Upon review of the data I would recommend the following improvements:

1.       Fig 2 experiment – block function of 4-1BB in supernatant to stop effect on splenocytes

2.       Figure 3D – there are no error bars suggesting only 1 experimental comparison – suggest at least 3 separate experiments.

3.       Fig 4B – lack error bars

Minor:

1.       proofreading, for example the test refers to figure 1D (line 231) but it means 2D

2.       Fig 3A – difficult to see OAdSA green line – recommend change for improved figure

 As the novelty of the manuscript is the addition of soluble 4-1BB peptide – one could enhance the conclusion by considering the following experiments for this or future work (although none of them would be required for current publication):

 Measuring 4-1BBL in blood of treated mice would further enhance the hypothesis of systemic activation

        Deactivating systemic 4-1BBL to decrease effect/immune cell infiltration would further support the hypothesis

        One could consider looking into LN activation of lymphocytes to demonstrate the systemic activation

Comments on the Quality of English Language

proofreading required for errors and gramma

Author Response

  1. Fig 2 experiment – block function of 4-1BB in supernatant to stop effect on splenocytes.

The control virus (OAdSA) has the same features as in OAdSA-4-1BBL except that it does not have 4-1BBL. As such we submit that this control suffices, and the data demonstrate the efficacy of SA-4-1BBL in driving the proliferation of T cells as reported by our group previously (1).

  1. Figure 3D – there are no error bars suggesting only 1 experimental comparison – suggest at least 3 separate experiments.

Thank you for the observation! We performed two additional experiments and a new graph for Fig. 3D is included showing the average three independent experiments within the revised manuscript’s version.

  1. Fig 4B – lack error bars

Fig. 4B is revised to include quantification of WBs from 3 independent experiments showing the media and standard deviation of the densitometry for the proteins LC3-I, LC-II and p62.

 Minor:

  1. proofreading, for example the test refers to figure 1D (line 231) but it means 2D.

Thank you; corrected.

  1. Fig 3A – difficult to see OAdSA green line – recommend change for improved figure. 

Thank you for noticing this lack of contrasting colors, we have updated the graphs with more distinguishable colors in all figures.

 As the novelty of the manuscript is the addition of soluble 4-1BB peptide – one could enhance the conclusion by considering the following experiments for this or future work (although none of them would be required for current publication):

Measuring 4-1BBL in blood of treated mice would further enhance the hypothesis of systemic activation.

Deactivating systemic 4-1BBL to decrease effect/immune cell infiltration would further support the hypothesis.

One could consider looking into LN activation of lymphocytes to demonstrate the systemic activation.

We appreciate the suggestions, which are incorporated into the Discussion section of the revised manuscript (p.13, Ln. 499-505).

Comments on the Quality of English Language

Revised manuscript is edited by a native English speaker.

  1. Schabowsky RH, Elpek KG, Madireddi S, Sharma RK, Yolcu ES, Bandura-Morgan L, et al. A novel form of 4-1BBL has better immunomodulatory activity than an agonistic anti-4-1BB Ab without Ab-associated severe toxicity. Vaccine. 2009;28(2):512-22.

Reviewer 2 Report

Comments and Suggestions for Authors

The authors developed an oncolytic adenovirus armed with 4-1BBL (OAdSA-4-1BBL) and examined its tumor suppressive potential using mouse lung cancer cells and a xenograft model. OAdSA-4-1BBL induced PCD I and II and autophagy-mediated ATP release. When OAdSA-4-1BBL was injected into the tumor, SA-4-1BBL protein was efficiently expressed in the tumor. It significantly suppressed the TC-1 mouse subcutaneous lung cancer, along with increased DC activation and CD8 T cell and NK cell infiltration into the tumor.

This has important implications for the development of oncolytic immunotherapy for lung cancer. However, there are some points that should be reconsidered.

The authors state that this OV primarily enhances the antitumor ability of CD8+ cytotoxic T cells against lung cancer. In Figure 2, the growth-promoting effect of conditioned medium supernatant of OAdSA-4-1BBL-infected TC-1 cells on splenocytes was examined. Oncolytic viruses may promote immune responses to both viral antigens and TAA. In this study, the authors used TC-1 cells and showed that HPV-16 E6/E7 is a potential target for cytotoxic T cells. To examine the effect of OAdSA-4-1BBL on antitumor immunity, but not on antiviral immunity, it is better to examine the immune response to E6/E7 of splenocytes from animals treated with OAdSA-4-1BBL.

Page 1, line 13: “Natural 4-1BBL(CD137)” may mean “Natural 4-1BB (CD137) ligand (4-1BBL) “.

Page 1, line 24: The authors state that “Tumor suppression was associated with the activation of dendritic cells — “-. Was the activation of DCs examined in this study?

Figure 1A: Is it possible that OAd expressing membrane type 4-1BBL induces oncolytic cell death, resulting in cell destruction, release of membrane type 4-1BBL, and stimulation of activated T cells?

Figure 5: These experiments were performed at an MOI of 50, and in vivo studies also appear to be performed at a high MOI. However, in a clinical setting, tumors are not necessarily infected at such high MOIs. Does OAdSA-4-1BBL replicate in tumor cells and spread viral infection to neighboring cells by progeny viruses? Have experiments been performed at lower MOIs? If so, does cell death proceed in a time-dependent manner?

Figure 7: The authors showed that OAdSA-4-1BBL infection causes changes in cell populations such as CD4+ T cells, CD8+ T cells, B cells, NK cells, and DCs in the tumor microenvironment. The proportion of each cell population in the total tumor-infiltrating immune cells in both infected and uninfected tumors needs to be described. Do these immune cells, especially DCs, allow expression of the SA-4-1BBL gene, even if they do not support productive replication?

Oncolytic viruses are currently being actively studied at the clinical level. How will the authors use this OAdSA-4-1BBL in the clinical setting to treat lung cancer? What is the delivery route? Comments on these are needed.

Author Response

The authors state that this OV primarily enhances the antitumor ability of CD8+ cytotoxic T cells against lung cancer. In Figure 2, the growth-promoting effect of conditioned medium supernatant of OAdSA-4-1BBL-infected TC-1 cells on splenocytes were examined. Oncolytic viruses may promote immune responses to both viral antigens and TAA. In this study, the authors used TC-1 cells and showed that HPV-16 E6/E7 is a potential target for cytotoxic T cells. To examine the effect of OAdSA-4-1BBL on antitumor immunity, but not on antiviral immunity, it is better to examine the immune response to E6/E7 of splenocytes from animals treated with OAdSA-4-1BBL.

We very much appreciate this insightful comment. Although we did not specifically investigate if tumor antigens other than E6/E7 are contributing to the efficacy of OAdSA-4-1BBL in this model, our group has extensively published on the ability of SA-4-1BBL to prime and expand CD8+ T cells to nonviral antigens, such as surviving, Sharma et al (1). Thus, we expect OAdSA-4-1BBL to generate T cells responses to both E6/E7 and nonviral antigens. We submit that a study specifically focusing on this issue is beyond the scope of this manuscript.

 Page 1, line 13: “Natural 4-1BBL(CD137)” may mean “Natural 4-1BB (CD137) ligand (4-1BBL) “.

Thank you for noting this. The text has been corrected accordingly using CD137L to refer to the ligand in parenthesis.

 Page 1, line 24: The authors state that “Tumor suppression was associated with the activation of dendritic cells — “-. Was the activation of DCs examined in this study?

Thank you for bringing this issue to our attention.  We did not specifically assess the activation of DC, rather their frequency within the tumor. The manuscript was modified to state “higher frequency” instead of activation.

Figure 1A: Is it possible that OAd expressing membrane type 4-1BBL induces oncolytic cell death, resulting in cell destruction, release of membrane type 4-1BBL, and stimulation of activated T cells?

Thank you for this insightful comment.  Indeed, this is a possibility that we discussed in the revised manuscript (p. 12; Ln. 457-461).

Figure 5: These experiments were performed at an MOI of 50, and in vivo studies also appear to be performed at a high MOI. However, in a clinical setting, tumors are not necessarily infected at such high MOIs. Does OAdSA-4-1BBL replicate in tumor cells and spread viral infection to neighboring cells by progeny viruses? Have experiments been performed at lower MOIs? If so, does cell death proceed in a time-dependent manner?

We consider this dose appropriate based on the presence of E1A as a replication marker for Ad homogenously distributed in a wide section of the tumor. The Ad E1A positive staining suggests that our OAds are actively infecting and replicating to the neighboring tumoral cells.  The dose we used is consistent with previous studies from our group where we treated melanoma subcutaneous tumors with intratumoral injections at the same dose/ifu of adenovirus (2).

Figure 7: The authors showed that OAdSA-4-1BBL infection causes changes in cell populations such as CD4+ T cells, CD8+ T cells, B cells, NK cells, and DCs in the tumor microenvironment. The proportion of each cell population in the total tumor-infiltrating immune cells in both infected and uninfected tumors needs to be described. Do these immune cells, especially DCs, allow expression of the SA-4-1BBL gene, even if they do not support productive replication?

Information for the total proportion of each type of cell from the total tumor-infiltrating immune cells (gating from CD45+) is now provided as a supplementary Fig 2 and supplementary table 2 and mentioned in the main text (p. 13; Ln. 500-503). Addressing the question if DCs can allow the expression of SA-1-1BBL, we didn’t perform any experiment in which we could isolate and detect the production of SA-4-1BBL from infected DCs.

Oncolytic viruses are currently being actively studied at the clinical level. How will the authors use this OAdSA-4-1BBL in the clinical setting to treat lung cancer? What is the delivery route? Comments on these are needed.

We suggest that the implementation of an inhaler device with a viral formulation to deliver the virus to the lungs, similar to the delivery of asthma drugs. Alternatively, a respiratory formulation containing the viral vectors can be nebulized to produce micro drops in saline solution with the use of a domestic nebulizer, which can be applied to the patient as an ambulatory therapy. This information was incorporated in the discussion section (p. 13; Ln. 510-514).

REFERENCES

  1. Sharma RK, Schabowsky RH, Srivastava AK, Elpek KG, Madireddi S, Zhao H, et al. 4-1BB ligand as an effective multifunctional immunomodulator and antigen delivery vehicle for the development of therapeutic cancer vaccines. Cancer Res. 2010;70(10):3945-54.
  2. Egger ME, McNally LR, Nitz J, McMasters KM, Gomez-Gutierrez JG. Adenovirus-mediated FKHRL1/TM sensitizes melanoma cells to apoptosis induced by temozolomide. Hum Gene Ther Clin Dev. 2014;25(3):186-95.

Round 2

Reviewer 1 Report

Comments and Suggestions for Authors

The manuscript relevant and provides innovative data contributing to the oncolytic virus field. The use of 4-1BBL is innovative.

one minor comment:

for the western quantification data, authors should provide statistical analysis.

Author Response

  1. For the western quantification data, authors should provide statistical analysis.

We added the information about the one-way ANOVA test we used to look for differences between groups, as well as the software information (Prism 10, GraphPad). In Materials and Methods section, page 4, line 149-151.